# Deep-Learning and Dynamic Time Warping-Based Approaches for the Diagnosis of Reactor Systems

**DOI:** 10.3390/s24237865

**Published:** 2024-12-09

**Authors:** Hoejun Jeong, Jihyun Kim, Doyun Jung, Jangwoo Kwon

**Affiliations:** 1Department of Electrical and Computer Engineering, Inha University, Incheon 22212, Republic of Korea; lilmae@inha.edu (H.J.); fuleaf99@inha.edu (J.K.); 2Korea Atomic Energy Research Institute, 111 Daedeok-daero 989-gil, Yusenong-gu, Daejeon 34057, Republic of Korea; jungdoyun@kaeri.re.kr; 3Department of Computer Engineering, Inha University, Incheon 22212, Republic of Korea

**Keywords:** deep learning, dynamic time warping, fault monitoring, fault diagnosis, reactor internals

## Abstract

The degradation of clamping force in the core support barrel, which forms the internal structure of a nuclear power plant, has the potential to significantly impact the plant’s safety and reliability. Previous studies have concentrated on the detection of clamping force degradation but have been constrained in their ability to identify the precise size and position. This study proposes a novel methodology for diagnosing the size and position of clamping force degradation in core support barrels, combining deep-learning techniques and dynamic time warping (DTW) algorithms. DTW is applied to the magnitude data of the ex-core neutron noise signal obtained in the frequency domain, thereby enabling the effective learning of changes in sensor data values. Moreover, autoencoder-based (AE-based) representation learning is utilized to extract features of the data, preventing overfitting and thus enhancing the robustness of the model. The experiment results demonstrate that the size and position of clamping force degradation can be accurately predicted. It is expected that this research will contribute to enhancing the precision and efficiency of internal structure monitoring in nuclear power plants.

## 1. Introduction

Nuclear power generation is a highly complex system consisting of a multitude of mechanical equipment and facilities. The failure of any single component can have a cascading effect, resulting in the failure of the entire system. Such an outcome could have considerable adverse effects on the environment, the economy, and human well-being. It is therefore evident that the early identification of minor irregularities and the accurate diagnosis and maintenance of the system prior to its complete failure can have considerable benefits for the operation of nuclear power plants. Moreover, this can have beneficial societal effects by improving public perception of the reliability of nuclear power generation.

A nuclear power plant comprises of a multitude of disparate components and diagnostic targets. The internal structure is the site of the nuclear power generation process and plays a major role in supporting and aligning the nuclear fuel and assemblies. From the design stage, efforts are made to ensure that the internal structures remain sound throughout the lifetime of the plant. Prior to startup, the internal structures are verified and operationalized through a comprehensive vibration assessment program. Among the internal structures, the hold-down springs, which are the mating components of the core support barrel (CSB) and the reactor pressure vessel, are a frequent area for defects. Therefore, numerous studies have been conducted to detect anomalies in these components.

This study proposes a methodology for diagnosing the position and size of weakening of the hold-down spring, extending beyond the detection of anomalies. In particular, the objective is to enhance diagnostic precision and extract high-dimensional features that are challenging for humans to analyze by employing deep learning, an artificial intelligence technology that has recently demonstrated remarkable performance in diverse domains. Moreover, the scarcity of training data represents a substantial obstacle, particularly given the distinctive characteristics of power plants and the heterogeneous nature of the input data across different sites. To address these issues, this study proposes a structure that is capable of learning only the change rate of data, rather than the sensor value itself, through the use of DTW-based data input conversion. Furthermore, it employs a methodology that distinguishes between representation learning and task learning, thus addressing the limitations of scarce training data and the issue of overfitting. To substantiate the proposed methodology, the accuracy of predicting the point and magnitude of the weakened clamping force was evaluated using data derived from physical analysis simulations. The results indicate that the proposed method can enhance the stability of nuclear power plants by demonstrating high accuracy.

The structure of this paper is as follows. First, a review of existing research on fault diagnosis for nuclear reactor internal structures is provided, introducing the application of deep-learning techniques in vibration analysis. Next, the Section 3 covers the dynamic time warping (DTW) algorithm and autoencoder-based representation learning proposed in this study. Following this, the dataset, experiment setup, and training and validation results of the model are presented. Finally, the conclusions summarize the research’s findings and suggest directions for future research.

## 2. Related Work

### 2.1. Diagnosing Anomalies in Nuclear Reactor Internals

The importance of nuclear reactor internals for nuclear power generation has been widely recognized, and numerous studies have examined the stability of these structures throughout the design, testing, and operational stages. As illustrated in Figure 1, a pressurized water reactor (PWR), the standard type in Korea, consists of a cylindrical vessel body joined by hemispherical upper and lower heads. The core support barrel, one of the main components of the internal structure, protects the nuclear fuel assemblies. It is located between the reactor upper head and the pressure vessel and is connected by hold-down springs. As illustrated in Figure 2, the upper head and hold-down springs are secured with 54 stud bolts. One of the most critical issues during reactor operation is the potential loss of clamping force in the core support barrel, often due to hold-down spring malfunctions [1]. A decrease in clamping force can lead to increased vibration, potentially compromising the protection of the nuclear fuel assemblies [2]. Therefore, monitoring clamping force degradation in the core support barrel is a key aspect of internal reactor monitoring.

Recent studies have applied deep-learning technologies for fault detection and diagnosis in nuclear reactor systems. These studies have been used to calibrate measurement systems, monitor the normal operation of subsystems, or assess the reactor core’s status. For instance, Lu et al. [3] utilized a two-layer MLP to predict thermohydraulic parameters critical for ensuring the normal operation of reactor systems. Ebrahimzadeh et al. [4] designed a neural network to detect faulty sensors within the reactor system and estimate correct values. The same study also employed an MLP to predict turbulent eddy viscosity inside the reactor and explored explainable-AI methodologies using SHAP. Furthermore, Rivas et al. [5] applied LSTM and CNN models to simultaneously detect anomalies in reactor systems and predict the remaining useful life (RUL). For monitoring the state of reactor internals specifically, recent works have employed data-driven methods with artificial neural networks. For example, Saleem et al. [6] proposed a method using three distinct DNNs to predict power peaking factors (PPFs), control rod bank levels, and cycle lengths. Additionally, Shriver et al. [7] developed a deep-learning approach to predict neutronic parameters in the 2D reflector fuel assemblies of pressurized water reactors (PWRs). These prior studies have largely focused on predictive models for parameters related to safety diagnostics, which assist in detecting core degradation or maintaining uniform power output. However, there has been little research on predicting the precise location and severity of faults in reactor internals. Therefore, this study proposes a deep-learning-based approach to detect clamping force anomalies and predict the specific position and size of degradation in the hold-down springs.

Changes in clamping force alter the vibration characteristics of core support barrels. Currently, nuclear power plants monitor the core support barrel’s vibration by analyzing the ex-core neutron detector’s noise signals during operation. This method of interpreting ex-core neutron signals dates back to F. de Hoffman’s 1946 observation of neutron fluctuations in reactor outputs [8,9,10,11]. Today, ex-core neutron noise signals, mainly used as inputs for core protection systems, are analyzed to monitor and diagnose internal structure vibrations [12,13,14]. Sensors, placed at 90-degree intervals at the top, upper, middle, and bottom of the reactor, capture a range of factors, including nuclear, thermohydraulic, mechanical, and damping effects.

Frequency analysis using the FFT technique is commonly applied to ex-core neutron noise signals, transforming the signals and calculating the autopower spectral density (APSD) [15]. The APSD is then analyzed across three vibration modes: beam, shell, and global. A rule-based system uses the peak frequency and magnitude changes in each mode to identify structural conditions. While effective, rule-based methods are limited, as they cannot provide detailed information on defect locations or types. Additionally, signal characteristics vary by power plant, installation, and operational conditions.

This study presents a novel methodology that uses advanced artificial intelligence to predict both the position and size of clamping force degradation in core support barrels. By employing deep learning, the proposed method addresses the limitations of current approaches and enhances diagnostic capabilities, ultimately contributing to the safety and reliability of nuclear power plant operations.

### 2.2. Vibration Analysis Using Deep-Learning

The application of artificial intelligence to vibration analysis has significantly advanced through numerous studies. With recent progress in deep-learning technology, research efforts are increasingly focused on using vibration data for condition diagnosis [16,17]. The use of deep-learning models in vibration analysis can be broadly classified by the input data type and the model architecture.

As shown in Figure 3, input data can be classified into three main types: time domain data [18,19,20], frequency domain data [21,22], and time–frequency domain data [23,24,25]. Time-domain data may be raw, time-varying sensor readings or sequences of features obtained through pre-processing or mechanical calculations. In the frequency domain, data are transformed using Fourier transform to capture magnitude and phase information. Time–frequency methods, such as short-time Fourier transform (STFT) [23,26,27], wavelet transform [24,28], and Hilbert–Huang transform [25,29], analyze data on both the time and frequency axes, often representing it as two-dimensional images for image analysis models.

In this study, we utilize frequency-domain data transformed by hardware FFT from ex-core neutron noise signals at nuclear power plant sites. To address varying mode characteristics across different plants, we analyze how frequency-domain data evolve compared to steady-state data.

Deep-learning methodologies can also be categorized by the feature extraction or backbone architecture used. Vibration signals, as sequential data, are well-suited for architectures like recurrent networks, convolutional neural networks [30], and self-attention mechanisms [31]. Recurrent networks analyze vibration data by processing sequences, as seen in models such as RNN [32], LSTM [33], and GRU [34]. Convolutional neural networks (CNNs) are designed to capture both local and global features by applying a kernel to the data. Wan et al. [35] improved the shortcomings of the traditional LeNet model to perform bearing fault diagnosis. Kim et al. [36] used grouped 1D convolution operations on three-axis vibration signals to extract features from each axis and classify faults. Li et al. [37] proposed a CNN model that takes vibration data in the time–frequency domain as input and effectively extracts both high-frequency and low-frequency information. Models such as WaveNet [38] and Temporal Convolutional Network (TCN) [39] are adapted for sequential data analysis. Zhan et al. [40] utilized a TCN-based model to extract the characteristics of time-series vibration data and detect abnormal vibrations in wind turbines. Self-attention mechanisms, especially Transformers [31], focus on relationships within the data by treating each segment as a “patch” and learning their interdependencies. Zhang et al. [41] applied an attention mechanism to the TCN model to effectively extract high-dimensional features and diagnose motor faults. Ahmed et al. [42] employed a transformer to capture the long-term temporal dependencies of vibration phenomena and performed bearing fault diagnosis.

For this study, we chose a CNN-based model as the backbone due to several advantages. Firstly, CNNs are effective for analyzing internal behaviors, which can be further enhanced with explainable-AI (XAI) [43] techniques for future diagnostic models. Secondly, frequency-domain analysis requires a focus on mode shape and size rather than on data sequence, where CNNs excel at extracting local features.

### 2.3. Similarity Measures Based on DTW

In data science, comparing two data sequences is a common task, with similarity defined as a measure of resemblance between them. Standard similarity measures include Euclidean distance, Manhattan distance, and cosine similarity. These methods typically calculate similarity on a point-by-point basis and then aggregate the results using an average or weighted sum. This approach assumes the points being compared are aligned on the same axis. For example, when comparing Data A and Data B, the algorithm evaluates corresponding points at each time, *t*, as shown in Figure 4. While these methods work well for assessing consistency across entire datasets, they are less effective for detecting dynamic changes. To address this, we apply the dynamic time warping (DTW) [44] algorithm, which allows us to compare data sequences with varying lengths, speeds, and feature alignments.

The DTW algorithm measures similarity between two sequences, even when they differ in length, speed, or feature alignment. As shown in Algorithm 1, the algorithm begins by constructing a cost matrix, *C*, of size n×m, where *n* and *m* are the lengths of the time series *X* and *Y*, respectively. Each cell in the matrix contains the Euclidean distance between corresponding points from the two series. Based on this cost matrix, a cumulative distance matrix is calculated, starting with the distance between $x_{1}$ and $y_{1}$ and progressing by choosing the minimum cumulative distance from neighboring cells. Finally, DTW identifies the point in *Y* with the smallest cumulative distance for each point in *X*.
**Algorithm 1** Dynamic Time Warping (DTW)**Require:** Two time series $X=(x_{1},x_{2},\cdots ,x_{n})$ and $Y=(y_{1},y_{2},\cdots ,y_{m})$

**Ensure:** DTW distance between *X* and *Y*
  1: Initialize cost matrix *D* of size n×m
  2: $D\left(1,1\right)\leftarrow d\left(x_{1},y_{1}\right)$
  3: **for** i←1,n **do**

  4:      **for** j←1,m **do**
  5:            $D\left(i,j\right)\leftarrow d\left(x_{i},y_{j}\right)+\min\left(D\left(i-1,j\right),D\left(i,j-1\right),D\left(i-1,j-1\right)\right)$

  6:      **end for**

  7: **end for**

  8: **return** D(n,m)           ▹ the DTW distance between *X* and *Y*

The DTW algorithm is commonly used for sequential data analysis in fields like speech recognition and natural language processing. In this study, we use DTW to organize input data in a way that captures and adapts to shifting patterns, enabling more accurate pattern recognition.

## 3. Methods

The proposed methodology is shown in Figure 5. To detect changes from the baseline state, a DTW-based input structure is used alongside an autoencoder (AE)-based representation learning method as the primary learning approach. A secondary regressor is then trained to refine the results. The following sections outline the specific characteristics of each component in this approach.

### 3.1. DTW-Based Input

In this study, the magnitude obtained from FFT processing of ex-core neutron noise signals is used as input to predict the clamping force of the core support barrel. This approach aligns with actual operating conditions in nuclear power plants, as their internal vibration monitoring systems (IVMSs) currently rely on frequency domain data.

Rather than using the original magnitudes directly, we first establish a baseline state and calculate changes relative to this baseline. This approach reflects the Comprehensive Vibration Assessment Program (CVAP) guidelines, which have shown that monitoring shifts and changes in vibration magnitudes is effective for detecting anomalies (NUCLEAR REGULATORY COMMISSION 1.20) [45]. Thus, changes in frequency can provide valuable information on potential anomalies. If the deep-learning model can learn complex change patterns that are difficult for humans to detect, it can accurately predict the clamping force status of the core support barrel. We chose DTW as the method for quantifying these changes, given its suitability for capturing shifts in patterns.

The reference state is defined as the steady state immediately after the power plant begins operation. Since real abnormal data are not available for this study, we use simulation data. In this simulation, data obtained under optimal clamping force conditions are designated as the reference data. The deep-learning model calculates the DTW distance between the reference data and new data. Instead of using DTW solely for distance calculation, we create a vector where each matched point has the normal data point as its origin and the new data point as its destination. This method allows for the inclusion of both the direction and magnitude of change in the training process. Figure 6 shows the difference between Euclidean distance and DTW vector comparison for normal and abnormal data. The utilized sensor data were collected from the eight sensors located at the top. In the plots, the lines in different colors represent the values from different sensors. The plot on the left visualizes the Euclidean distances of the sensor values in a normal state, while the plot on the right illustrates the matched vectors using dynamic time warping (DTW). By using DTW-based change rates as input, the algorithm can assess frequency changes even when baseline conditions vary, ensuring robust performance.

### 3.2. AE-Based Representation Learning

The unique characteristics of nuclear power plants make it challenging to obtain anomalous data, and simulation-based data collection is also limited by the inherent constraints of simulations. Consequently, the deep-learning model is at risk of overfitting. To mitigate this, we propose a two-stage learning approach, consisting of a distinct representation learning phase followed by task learning with a shallow predictive model. This two-stage approach is depicted in Figure 7.

Typically, deep-learning methods utilize pre-trained models for image or time series data. However, the unconventional nature of our data prevents the use of standard backbone models, requiring us to build the model from scratch. Training a deep model with limited data is challenging; hence, we use an autoencoder [46] for representation learning. Prior studies have shown that autoencoders are effective in extracting meaningful features.

For this study, we first built a CNN-based autoencoder and conducted preliminary training to ensure that the model could restore the input DTW data and capture essential features. Each convolutional layer is modified to use 1D convolutions to better handle sequential data. Further details of the model implementation are covered in the Section 4.

Once the autoencoder’s representation training is complete, we utilize the encoder as the feature extractor. To prevent overfitting and achieve robust learning, a shallow neural network model is used to predict the center point and extent of the weakened clamping force in the core support barrel, using features extracted by the encoder. Task learning is then performed solely on the regressor, which is a shallow network, while the encoder weights remain fixed. This ensures that only information from the initial restoration training contributes to the extracted features, thus avoiding overfitting on the test data. This structure is advantageous when there is a wide range of possible anomaly cases (position and size variations) but limited data per case. The primary objective of this approach is to enhance robustness for diverse applications across different power plants.

## 4. Experiment and Results Analysis

### 4.1. Dataset

To validate the proposed methodology, a specific experimental scenario was designed. First, the weakened clamping force area was determined based on the 54 bolts connecting the core support barrel to the hold-down spring. This setup allows the measurement area of the clamping force to be represented by 54 discrete sections, rather than a continuous 360-degree space. The second assumption is that the weakening of clamping force occurs in a single area. In practice, multiple areas may experience reduced clamping force simultaneously; however, for this experiment we assume that the weakening occurs in a single contiguous section, regardless of the weakened area’s size.

The simulated data were generated using a finite element analysis (FEA) model developed by the Korea Atomic Energy Research Institute. This model simulates and analyzes the abnormal vibration data of internal reactor structures at full scale. The FEA model calculates the support stiffness between the core support structure, reactor vessel, and hold-down ring, replicating the environmental conditions of an actual reactor vessel in a steady state. The model’s behavior was verified to closely resemble that of a commercial reactor under steady-state conditions [47]. To simulate various failure conditions, the clamping force on the stud bolts was set between 0 (completely loosened) and 1 (fully tightened).

The data obtained from each simulation were considered as single data points, giving a total of 6575 data points. Each individual data point has a different hole size, hole center, and clamping force for the bolt. The hole size refers to the number of points in the hole where there is a zone of reduced clamp load, while the hole center refers to the center point of the zone. In this study, we are not predicting the clamping force, only the hole size and hole center. A total of 107 locations were considered, with hole sizes ranging from 0 to 6, with even hole sizes resulting in a value between the two measurement ranges.

For each case, data were collected from eight sensors positioned at the top, upper, middle, and bottom, resulting in 32 channels. Each channel contains phase and magnitude values; however, only magnitude values were used as inputs in this study, as discussed in Section 4.3.1. Each magnitude value is represented on a frequency axis with intervals of 0.3, covering the range from 0 to 30.

To ensure consistency with real-world conditions, the dataset utilized in this study comprises data stored in the frequency domain, specifically transformed using fast Fourier transform (FFT). This approach reflects the fact that data collected from actual nuclear reactor facilities are typically pre-processed into the frequency domain and not stored in their raw time-series format. Moreover, the simulation data generated for this study follows the same pre-processing pipeline, thereby maintaining consistency between simulation data and real-world data. Consequently, employing frequency-domain inputs aligns with the characteristics of the data available for practical applications. This ensures that the methodology developed in this study can be seamlessly applied to real-world monitoring scenarios without requiring additional transformation processes.

### 4.2. Experiment Setting

To validate the proposed methodology, we conducted three main experiments. First, we analyzed the effects of using magnitude and phase as inputs. We assessed performance using three configurations: magnitude alone, phase alone, and both combined.

Secondly, we evaluated the effectiveness of using the difference between the input signal and the reference signal as input, rather than the original sensed value. We hypothesize that this difference-based input, particularly when processed with DTW, provides superior learning outcomes compared to the original signal or other methods, such as Euclidean distance. To demonstrate this, we analyzed how these three inputs (original signal, Euclidean distance, and DTW vector) influence learning and performance.

Thirdly, we examined the impact of AE-based representation learning on model robustness. We claim that representation learning leads to more resilient models. To verify this, we compared different methods: one that applies representation learning followed by training on a shallow network and another that trains the model from scratch without representation learning. These comparisons were conducted across data processed with original values, Euclidean distances, and DTW vectors.

The model used in our experiments comprises two main components. The first is an autoencoder (AE) for feature extraction and representation learning, and the second is a regressor for making predictions based on these features. As shown in Figure 8, the AE model on the left includes three blocks: a Conv block, an Encoder block, and a Decoder block. The Conv block extracts features while maintaining the length and dimensionality of the input data. It performs a 1D convolution with a kernel size of 3, stride 1, and padding 1, followed by batch normalization and ReLU activation. The Encoder block reduces the length and doubles the dimensionality using a convolution of length 2, stride 2, with ReLU activation. In the Decoder block, we use a deconvolution operation to double the length and halve the dimensionality, then concatenate this output with the tensor from the Encoder block. Batch normalization and a final Conv block further refine the features. The Encoder and Decoder blocks are designed to perform complementary operations.

The AE model has three layers in both the encoder and decoder. The first encoder layer and the last decoder layer adjust channel sizes between the input and 64 channels. Two additional Conv blocks are inserted between the encoder and decoder for further feature refinement, resulting in a flat-layer architecture.

The regressor model, used to predict the position and size of clamping force degradation, includes two convolutional layers and three fully connected layers. This setup maximizes the benefits of AE-based representation learning and mitigates overfitting risks that could arise from a larger model, as the input data are already well-characterized by the use of DTW vectors.

### 4.3. Experiment Result

#### 4.3.1. Comparison of Magnitude and Phase Suitability

In this study, we evaluated the effectiveness of using phase and magnitude from the frequency domain of ex-core neutron noise signal data. The experiments were divided into three cases: phase only, magnitude only, and both combined. We analyzed the results of the proposed method without applying representation learning to assess the impact of each type of input.

Magnitude represents the energy distribution within the frequency spectrum and is used to identify frequency-based anomaly patterns. Phase, on the other hand, reflects the relative temporal relationships between frequency components, offering dynamic insights into the actual operational conditions of the equipment. By combining magnitude and phase data, a more holistic perspective is achieved, capturing both changes in energy patterns at specific frequencies and dynamic behaviors. This study systematically investigates the impact of each input type on the model’s diagnostic performance through a series of experiments, aiming to identify the most effective input configuration. Additionally, to examine the effectiveness of the DTW-based sensor value change learning approach, we conducted experiments using either raw sensor values or DTW vectors as inputs. To quantitatively compare the performance of predicting the position and size of clamping force degradation, we used a threshold similar to those used in classification tasks. Predictions were deemed correct if the difference between the model’s result and the actual value fell below the threshold; otherwise, they were considered incorrect. In Table 1, “Threshold” refers to the accuracy threshold, “Value” indicates the use of raw sensor values as input, and “DTW” indicates the use of DTW vectors.

Our experiments revealed that accuracy was higher when using DTW vectors derived from magnitude data compared to raw sensor values. This suggests that DTW is less suitable for phase data, as shifts in phase do not correspond to similar patterns in the data, making them difficult to compare. Consequently, we selected magnitude data as the input for the DTW-based method proposed in this study. All subsequent experiments are based on magnitude inputs.

#### 4.3.2. Effect of the DTW Algorithm

In this study, we analyzed the effectiveness of using DTW to capture changes in sensor values relative to reference data, rather than relying on raw sensor values alone. We conducted training and validation using three different input methods: the first method used the raw magnitude values as input, while the other two used the change from the reference data calculated with Euclidean distance and DTW vectors, respectively.

Training and validation losses for each input method are shown in Figure 9, where the blue graph represents training loss and the red graph represents validation loss.

The experimental results indicate that validation loss varies less when predicting the position of clamping force degradation compared to predicting its overall size. This suggests that position prediction is less susceptible to overfitting, leading to more stable learning. For the validation loss, DTW exhibits the least variation, demonstrating stability and quick convergence, which supports the superiority of DTW in the learning process.

Table 2 presents the test results for both tasks. As in previous experiments, accuracy was calculated based on specified thresholds. From Table 2, it is evident that using DTW for training enhances model accuracy for both tasks. For size prediction, accuracy improved by 200% (from 0.28 to 0.84) compared to using raw sensor values, while position prediction accuracy increased by 84% (from 0.13 to 0.24).

#### 4.3.3. Effect of AE-Based Representation Learning

In this study, we assessed the effectiveness of the proposed AE-based representation learning, followed by downstream task learning. Using the same setup as the previous experiment on input effects, we conducted six training and validation experiments. These included an experimental group that applied AE-based pre-training for each of the three inputs and another group that did not apply pre-training. Figure 10 and Figure 11 show the changes in loss for both groups. Figure 10 displays the loss reduction in predicting preload size, while Figure 11 shows the loss reduction in predicting preload location. In both graphs, the red line represents validation loss and the blue line represents training loss.

The results indicate that, in most cases, prior representation learning leads to a greater reduction in validation loss and a narrower range of oscillation, particularly during convergence. This suggests that representation learning effectively helps to prevent overfitting.

Table 2 shows the accuracy measures for both tasks after training. As in the previous experiment, we used threshold-based accuracy. In Table 3, “Scratch” refers to results from training without representation learning, while “PreTrain” refers to results from pre-training with representation learning. The findings show that pre-training with representation learning improves accuracy, especially in cases using Euclidean distance and raw signal inputs, which initially had lower accuracy. These accuracies improved by 25% and 23%, respectively.

### 4.4. Benchmark

To evaluate the accuracy and performance of the proposed model, a benchmark was conducted. The benchmark models included MLP, CNN, and Transformer. For fair comparison with the proposed model, all benchmark models were designed as networks with four layers, including the final prediction layer. The same activation function (ReLU) and regularization technique (Batch Normalization) were applied, and all models were trained for 500 epochs before performance evaluation. Training and validation for all models were conducted under the same computing environment, as detailed in Table 4 below.

The following Table 5 presents benchmark results comparing MLP, CNN, Transformer, and the proposed model. The comparison metrics include Number of Parameters, Model Parameter Size, Inference Time, Accuracy on Size Prediction, and Accuracy on Position Prediction. An analysis of the results based on each metric is provided below.

The number of parameters and model parameter size reflect the memory requirements for model operation, where smaller values indicate more compact models, while larger values imply higher model capacity. The MLP is the largest model, with 26.4 M parameters and a parameter size of 105.7 MB. On the other hand, the CNN is the smallest model, with 1.6M parameters and a parameter size of 6.514 MB. The Transformer and the proposed model fall into the mid-range, demonstrating a balance between efficiency and capacity.

Inference time was measured in seconds, representing the time required for each model to perform a single inference under the specified computing resources. The Transformer exhibited the fastest inference time due to the computational efficiency of its self-attention mechanism, which has fewer operations relative to its memory demands. The proposed model showed the slowest inference time at 0.099 s. However, this speed is still sufficient to meet real-time requirements in practical applications.

Lastly, the accuracy of the models was analyzed with a threshold of 0. The proposed model achieved the highest accuracy in both size prediction (0.83) and position prediction (0.31). While the CNN performed second-best in size prediction, it showed poor performance in position prediction. Conversely, the MLP demonstrated relatively better performance in position prediction but underperformed in size prediction. These results confirm that the proposed model outperforms the benchmark models in terms of predictive accuracy for both tasks.

In conclusion, while the proposed model is slightly slower in terms of inference time compared to the benchmarks, it demonstrates superior performance in model capacity and predictive accuracy. Furthermore, with an inference time of 0.099 s, allowing approximately 100 inferences per second, the proposed model is expected to be suitable for real-time applications in field environments.

## 5. Conclusions

In this study, we proposed a method that leverages deep learning to identify the precise position and extent of core support barrel degradation within nuclear power plant internal structures, rather than simply detecting its presence. The proposed approach incorporates dynamic time warping (DTW) to address data variability across different plant sites and efficiently extracts valuable features through autoencoder-based pre-training. To verify the method’s effectiveness, we trained and validated the model using simulation data provided by the Korea Atomic Energy Research Institute. An ablation study compared the performance of each module, showing that DTW achieved accuracy improvements of up to 2× for position prediction and 3× for size prediction compared to using raw sensor data alone. For AE-based representation learning, performance improvements of up to 33%, 23%, and 31% were observed for sensor inputs, Euclidean distance, and DTW, respectively, in the challenging position prediction task. In all cases, we quantitatively demonstrated that the proposed method enhances performance compared to the baseline.

In this study, we limited validation to simulation data due to the challenges of accessing real power plant data. To ensure robustness within the simulation domain, we employed a DTW-based approach that emphasizes variations, allowing the model to remain invariant to simulation-specific characteristics. Future research will focus on validating the method with real-world data or data from experimental setups that closely mimic actual conditions. Additionally, incorporating technologies such as explainable AI (XAI) could enhance the interpretability of AI predictions, further supporting practical applications. Furthermore, domain adaptation techniques will be explored to bridge the gap between simulation data and real-world environments, enabling broader applicability of the proposed methodology.

## Figures and Tables

**Figure 1 sensors-24-07865-f001:**
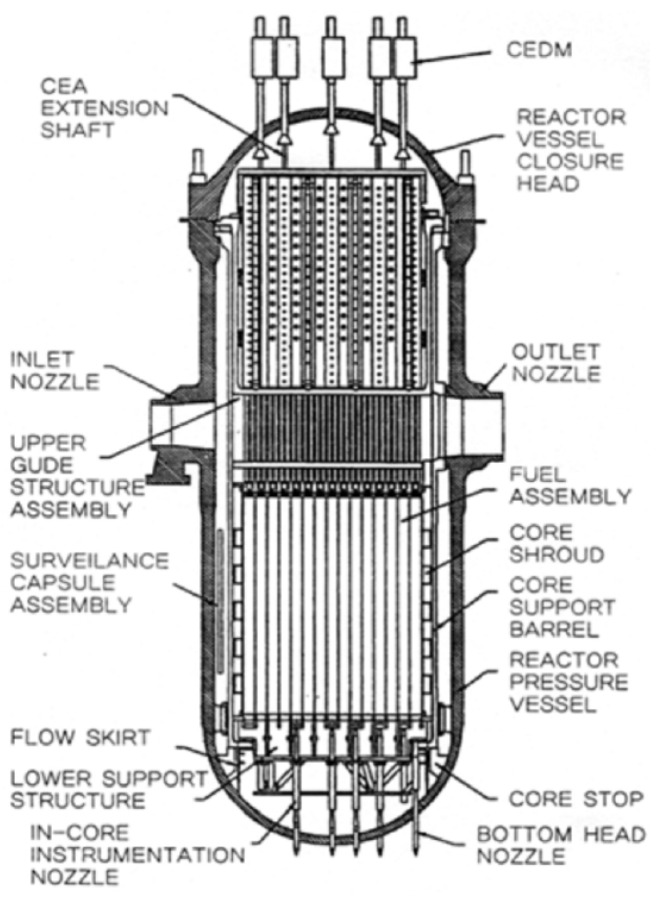
Components of the internal structure of the reactor.

**Figure 2 sensors-24-07865-f002:**
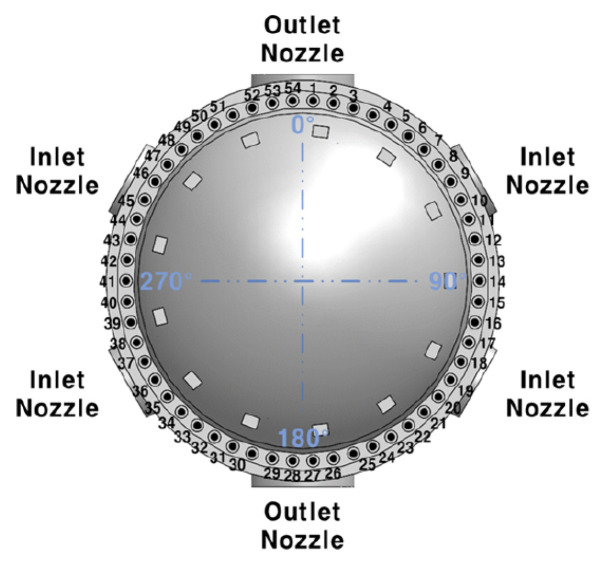
A top-down view of the reactor’s internal structure. It is firmly supported and secured by 54 stud bolts.

**Figure 3 sensors-24-07865-f003:**
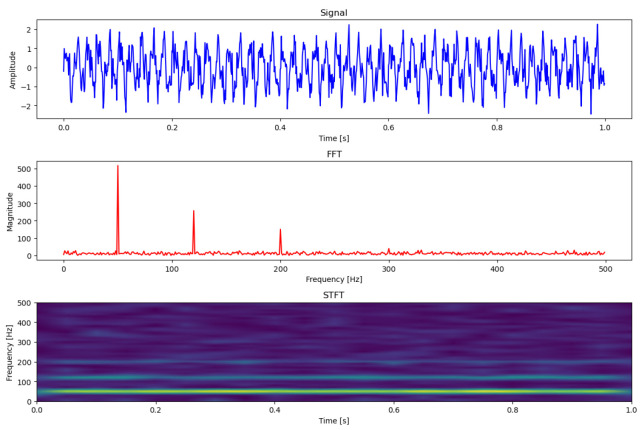
Comparison of each domain of vibration input in deep learning (top: signal domain, middle: frequency domain, bottom: time–frequency domain).

**Figure 4 sensors-24-07865-f004:**
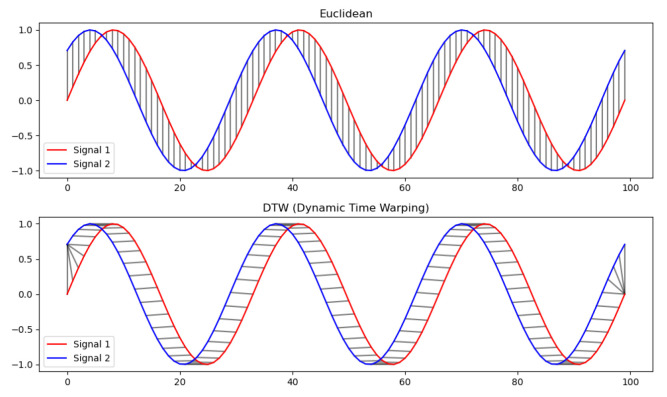
Example of distance comparison using Euclidean distance and DTW algorithm.

**Figure 5 sensors-24-07865-f005:**
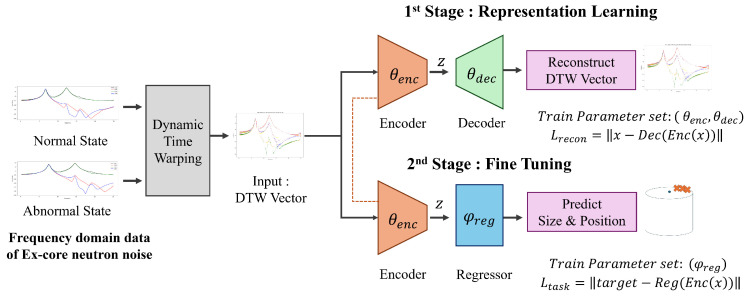
The proposed model architecture. θ and ϕ represent the learnable parameters of the model. $\theta_{\mathrm{enc}}$ denotes the parameters of the encoder and $\theta_{\mathrm{dec}}$ denotes the parameters of the decoder. These parameters are optimized during the first stage (Representation Learning) to encode the input data into a latent space and reconstruct it. On the other hand, $\varphi_{\mathrm{reg}}$ represents the learnable parameters of the regression model, which is used in the second stage (Fine Tuning) to predict the size and position. In the Fine Tuning stage, only $\varphi_{\mathrm{reg}}$ is updated, while $\theta_{\mathrm{enc}}$ remain frozen.

**Figure 6 sensors-24-07865-f006:**
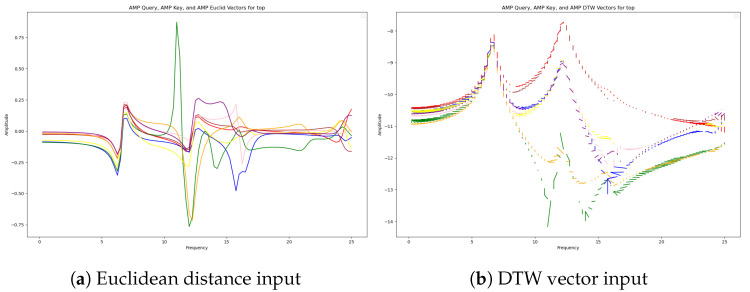
Difference between input methods using DTW vector and Euclidean distance.

**Figure 7 sensors-24-07865-f007:**
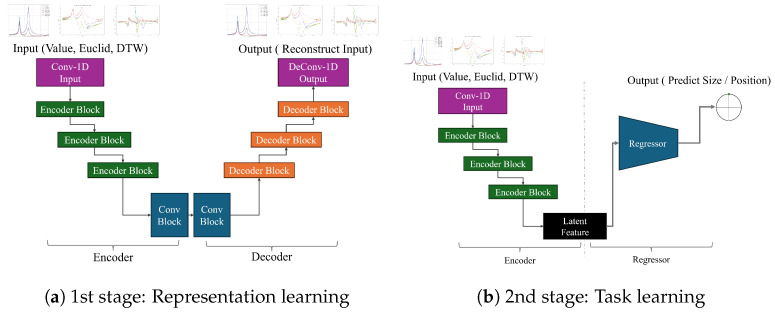
Two-step learning structure consisting of representation learning and regression.

**Figure 8 sensors-24-07865-f008:**
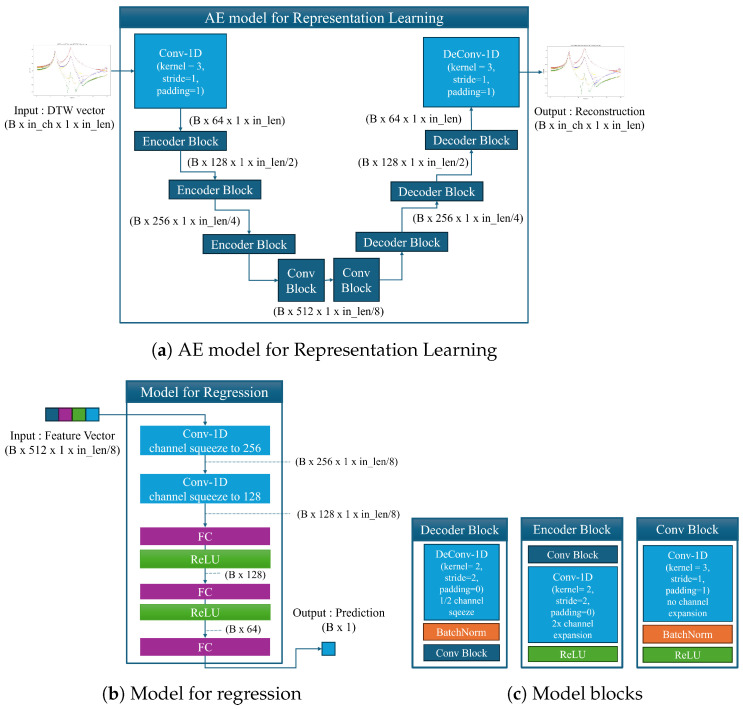
Model structure.

**Figure 9 sensors-24-07865-f009:**
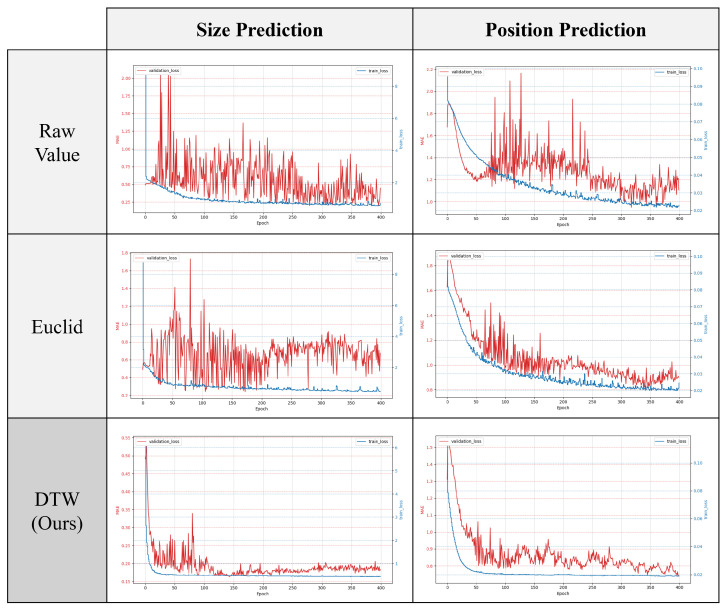
Comparison of training and validation losses according to input method.

**Figure 10 sensors-24-07865-f010:**
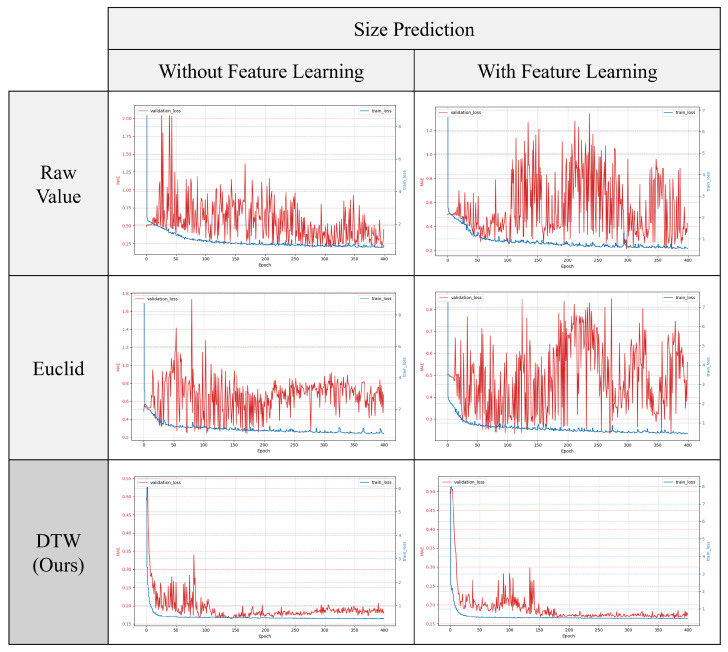
Comparison of training and validation losses with and without representation learning in size prediction.

**Figure 11 sensors-24-07865-f011:**
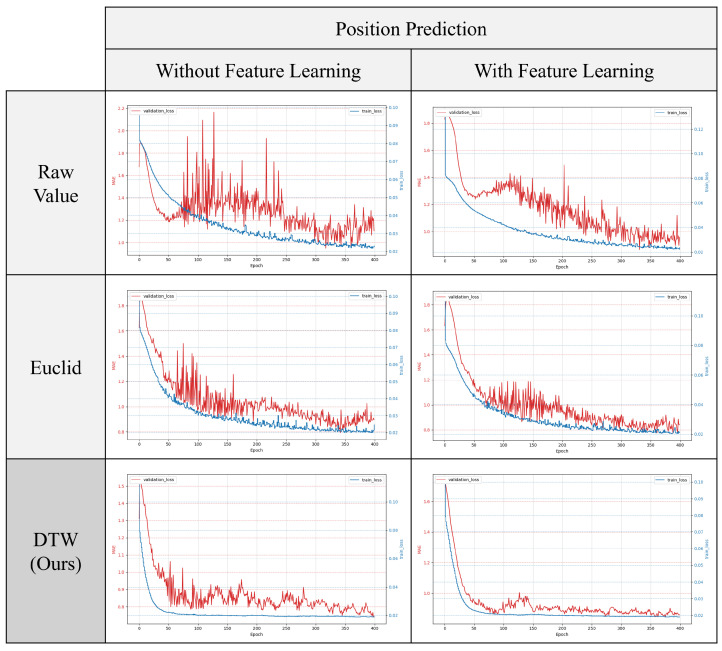
Comparison of training and validation losses with and without representation learning in position prediction.

**Table 1 sensors-24-07865-t001:** Accuracy by input data.

	Size Prediction						Position Prediction					
	Value			DTW			Value			DTW		
Thres	Amp	Phase	Both	Amp	Phase	Both	Amp	Phase	Both	Amp	Phase	Both
**0**	0.28	0.75	0.55	0.84	0.65	0.76	0.13	0.18	0.24	0.24	0.10	0.13
**1**	0.53	0.92	0.91	0.93	0.96	0.91	0.31	0.32	0.41	0.45	0.15	0.21
**2**	0.81	0.98	0.99	0.99	1	0.99	0.46	0.47	0.48	0.60	0.24	0.27
**3**	0.98	1	1	1	1	1	0.63	0.56	0.55	0.71	0.31	0.34
**4**	1	1	1	1	1	1	0.68	0.61	0.6	0.75	0.4	0.42

**Table 2 sensors-24-07865-t002:** Accuracy for different shapes of amplitude data.

	Size Prediction			Position Prediction		
Threshold	Value	Euclid	DTW (Ours)	Value	Euclid	DTW (Ours)
**0**	0.28	0.6	**0.84**	0.13	0.07	**0.24**
**1**	0.53	0.77	**0.93**	0.30	0.12	**0.45**
**2**	0.81	0.92	**0.99**	0.46	0.25	**0.60**
**3**	0.98	1	**1**	0.63	0.4	**0.71**
**4**	1	1	**1**	0.68	0.52	**0.75**

**Table 3 sensors-24-07865-t003:** Accuracy by feature learning.

	Size Prediction						Position Prediction					
	Value		Euclid		DTW		Value		Euclid		DTW	
Thres	Scratch	PreTrain	Scratch	PreTrain	Scratch	PreTrain (Ours)	Scratch	PreTrain	Scratch	PreTrain	Scratch	PreTrain (Ours)
**0**	0.28	0.35	0.6	0.63	0.84	**0.83**	0.13	0.16	0.07	0.21	0.24	**0.31**
**1**	0.53	0.66	0.77	0.84	0.93	**0.93**	0.30	0.40	0.12	0.38	0.45	**0.51**
**2**	0.81	0.91	0.92	0.98	0.99	**0.99**	0.46	0.59	0.25	0.51	0.60	**0.62**
**3**	0.98	1	1	0.99	1	**1**	0.63	0.66	0.40	0.62	0.71	**0.74**
**4**	1	1	1	1	1	**1**	0.68	0.69	0.52	0.69	0.75	**0.77**

**Table 4 sensors-24-07865-t004:** Specifications of the system used for the experiments.

Device	Specifications
CPU	Intel(R) Core(TM) i9-10900 CPU @ 2.80 GHz
RAM	128 GB
GPU	NVIDIA GeForce RTX 4090
OS	Ubuntu 22.04 LTS
Python	3.10

**Table 5 sensors-24-07865-t005:** Performance comparison of different models in terms of the number of parameters, model parameter size, inference time, accuracy on size prediction, and accuracy on position prediction.

	MLP	CNN	Transformer	Proposed Model
Number of Parameters	26.4 M	1.6 M	4.3 M	4.8 M
Model Param Size	105.7 MB	6.514 MB	17.310 MB	19.3 MB
Inference Time	0.023	0.092	0.005	0.099
Accuracy on Size Prediction	0.39	0.71	0.38	0.83
Accuracy on Position Prediction	0.17	0.04	0.05	0.31

## Data Availability

The datasets presented in this article are not readily available due to issues related to copyright.

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
