# Peer review of "Deep-Learning and Dynamic Time Warping-Based Approaches for the Diagnosis of Reactor Systems"

_sensors, 2024, doi:10.3390/s24237865_

Round 1
Reviewer 1 Report
Comments and Suggestions for Authors
In this manuscript, the authors propose a novel methodology for diagnosing the size and position of clamping force degradation in core support barrels, combining deep learning techniques and dynamic time warping algorithms. The experimental results demonstrate that the size and position of clamping force degradation can be accurately predicted.
There are some concerns that need to be addressed as follows:
1. It is suggested to conduct experimental comparisons between the proposed method and other advanced methods.
2. There are some problems in the format of references. It is suggested to add references in recent three years, and the following articles may help to strengthen this manuscript.
[1] Rolling-element bearing fault diagnosis using improved LeNet-5 network
3. The English writing needs to be improved greatly.
Comments on the Quality of English LanguageNeeded to be improved.
Author Response
Dear Reviewer,
We would like to express our sincere gratitude for your insightful comments and suggestions, which have been invaluable in improving the quality of our manuscript. Below, we address each of your comments in detail:
Comment 1:
It is suggested to conduct experimental comparisons between the proposed method and other advanced methods.
Response 1:
We appreciate your suggestion to include experimental comparisons with other advanced methods. In response, we have added a new subsection, Section 4.4: Benchmark, where we compare the proposed method with models such as MLP, CNN, and Transformer. This addition provides a clearer understanding of the advantages of our approach. The details can be found on page 13, starting from line 392.
Comment 2:
There are some problems in the format of references. It is suggested to add references in recent three years, and the following articles may help to strengthen this manuscript.
Response 2:
Thank you for pointing out the issue with the references and for suggesting a relevant article. We have incorporated the recommended reference along with additional recent studies to strengthen the manuscript. These updates are reflected in Section 2.2: Vibration Analysis using Deep Learning. The revised content begins on page 4, line 140.
Comment 3:
The English writing needs to be improved greatly.
Response 3:
To address your concern about the quality of English writing, we have carefully revised the manuscript to improve grammar, terminology, and overall readability. We believe these revisions have significantly enhanced the clarity and professionalism of the paper.
Once again, we deeply appreciate your valuable feedback, which has been instrumental in refining our work.
Reviewer 2 Report
Comments and Suggestions for Authors
The paper deals with the suggestion of a method using deep learning and Dynamic Time Warping (DTW) to identify the precise position and extent of core support barrel degradation within nuclear power plant internal structures, rather than simply detecting its presence.
The authors have verified the effectiveness of their suggested method by training and validating the proposed prediction model through simulation works and model based data provided by the Korean Atomic Energy Research Institute. The problem is stated as a vibration monitoring based study using Artificial intelligence (CNN).
- Despite the interseting results, it is nevertheless necessary to take into account the following remarks:
- The authors have claimed that they used experimental data, but as mentionned in section 4 and in the conclusion, the used data are obtained by a simulated model using Finite Element. Thus the term "experimental" must be modified.
- The figures from 5 to 11 must be enhanced for a good reading.
- The authors have only considered the accuracy of the results but they have not considered the problem of the algorithmic complexity wich influences considerably the time machine (the execution time).
- The references are not presented correctly, especially the references: 4, 7, 10, 13, from 15 to 18. 20 , 22, 23, 24, 26, 30 and 37.The authors have to correct them according to the instructions for authors of the journal.
-
-
Author Response
Dear Reviewer,
We sincerely appreciate your thorough review and constructive feedback on our manuscript. Your comments have been invaluable in improving the quality and clarity of our work. Below, we address each of your concerns in detail:
Comment 1:
- The authors have claimed that they used experimental data, but as mentionned in section 4 and in the conclusion, the used data are obtained by a simulated model using Finite Element. Thus the term "experimental" must be modified.
Response 1:
We understand your concern regarding the use of the term “experimental” to describe the data obtained through simulations. In response, we have replaced all instances of “experimental” with “simulated” throughout the manuscript to ensure consistency and accuracy.
Comment 2:
- The figures from 5 to 11 must be enhanced for a good reading.
Response 2:
To improve the readability of the figures, we have made several adjustments. Specifically, we have increased the size of key figures, such as those presenting experimental results (Figures 9–11) and model structures (Figures 5–8). Additionally, unnecessary details have been removed to enhance clarity and focus on the critical information.
Comment 3:
- The authors have only considered the accuracy of the results but they have not considered the problem of the algorithmic complexity wich influences considerably the time machine (the execution time).
Response 3:
In response to your suggestion to consider algorithmic complexity and execution time, we have included a new subsection, Section 4.4: Benchmark. This section compares the proposed model with other models, such as MLP, CNN, and Transformer, not only in terms of performance but also execution time. The details of this comparison can be found on page 13, starting from line 392.
Comment 4:
- The references are not presented correctly, especially the references: 4, 7, 10, 13, from 15 to 18. 20 , 22, 23, 24, 26, 30 and 37.The authors have to correct them according to the instructions for authors of the journal.
Response 4:
We have carefully reviewed and corrected the formatting of the references you identified (4, 7, 10, 13, 15–18, 20, 22–24, 26, 30, and 37) to align with the journal’s instructions for authors.
Once again, we deeply appreciate your valuable feedback, which has significantly improved the quality of our manuscript.
Reviewer 3 Report
Comments and Suggestions for Authors
2.2. Vibration Analysis using Deep Learning:
There are already some deep learning-based diagnostic methods for reactor systems. However, in this section, the author only introduces general deep learning methods and lacks a detailed discussion of existing deep learning-based reactor system diagnostic methods. In particular, the analysis of the strengths and weaknesses of these existing methods is insufficient. The author should supplement this with a review of related studies, clearly outlining the limitations of existing methods and areas for improvement to strengthen the innovation and background discussion of the paper.
2.2. Vibration Analysis using Deep Learning:
The author mentions the selection of a CNN model for the study but does not further explain why the ResNet variant was chosen specifically. Have other CNN variants (e.g., VGG, Inception) been considered? The author should provide an analysis of the advantages and disadvantages of different CNN variants and clarify the reasons for choosing ResNet to enhance the theoretical foundation of the research.
3.1. DTW-based Input:
The author has not provided a detailed explanation of why the vibration monitoring system (IVMS) currently relies on frequency-domain data input. What are the advantages of using frequency-domain data input compared to time-domain or time-frequency-domain methods? Moreover, considering that time-frequency analysis methods (e.g., STFT, Hilbert-Huang transform) could also be applied to this problem, the author should explain why FFT was chosen as the frequency-domain analysis method.
3.1. DTW-based Input:
The symbols used in Figure 5 (e.g., /theta_{enc} and /theta_{dec}) are not explained, and I did not see any related formula derivations or symbol definitions. To improve the readability and understanding of the figure, the author should provide further details on Figure 5 and explain the relevant symbols and their physical meanings.
3.1. DTW-based Input:
In Figure 6, the author mentions that it shows the difference between Euclidean distance and DTW (Dynamic Time Warping) comparisons for normal and abnormal data. However, the meaning of each curve in the two subplots is unclear. The figure lacks a legend, and there is no explanation of whether different colors have specific meanings. Additionally, there is no further discussion of Figure 6 in the subsequent text, which leaves me confused about the message the author intends to convey. I recommend the author provide a more detailed explanation of Figure 6 and, if necessary, update the figure to make it more understandable. Moreover, the text in the figure is too small; I suggest increasing the font size to improve readability.
4.1 Dataset:
There are two repetitive paragraphs in this section. Is there any specific reason for this arrangement? The author should clarify this part to help readers understand.
4.2 Experiment Setting:
The author evaluates model performance using three configurations: amplitude alone, phase alone, and a combination of both. Why were these three configurations chosen? The author needs to explain the rationale for selecting these configurations and specify whether other input features were considered for evaluating the network performance. If necessary, please add experiments to validate the impact of different input configurations on model performance.
4. Experiment and Results Analysis:
The author mentions training and validation with simulated data, as real data could not be obtained, as stated in the introduction. However, there is typically a large discrepancy between simulated data and real experimental signals. The author should further explain why the training results obtained with simulated data can be used for fault diagnosis of real experimental signals. If normal signal data from a real environment is available, could the author include tests on real normal signals to address concerns about the neural network trained on simulated data's ability to diagnose real signals?
Comments on the Quality of English LanguageThe English quality is quite good, but some sentences are excessively lengthy.
Author Response
Dear Reviewer,
We sincerely appreciate your thorough and insightful feedback, which has been instrumental in improving the quality and clarity of our manuscript. Below, we provide responses to each of your comments:
Comment 1:
2.2. Vibration Analysis using Deep Learning:
There are already some deep learning-based diagnostic methods for reactor systems. However, in this section, the author only introduces general deep learning methods and lacks a detailed discussion of existing deep learning-based reactor system diagnostic methods. In particular, the analysis of the strengths and weaknesses of these existing methods is insufficient. The author should supplement this with a review of related studies, clearly outlining the limitations of existing methods and areas for improvement to strengthen the innovation and background discussion of the paper.
Response 1:
We conducted additional literature research and included discussions on studies that utilize deep learning for reactor diagnostics. These updates provide a detailed analysis of the strengths and weaknesses of existing methods, as well as the limitations they aim to address. We have explicitly highlighted how our approach differs from these studies, emphasizing the originality of our work. The revised content can be found in Section 2.2: Vibration Analysis using Deep Learning.
Comment 2:
2.2. Vibration Analysis using Deep Learning:
The author mentions the selection of a CNN model for the study but does not further explain why the ResNet variant was chosen specifically. Have other CNN variants (e.g., VGG, Inception) been considered? The author should provide an analysis of the advantages and disadvantages of different CNN variants and clarify the reasons for choosing ResNet to enhance the theoretical foundation of the research.
Response 2:
We apologize for the confusion caused by the mention of ResNet. Our intention was to reference specific structural characteristics proposed in ResNet rather than the residual operations. To avoid further misunderstanding, we have removed references to ResNet throughout the manuscript and clarified that our study is based on a CNN backbone.
Comment 3:
3.1. DTW-based Input:
The author has not provided a detailed explanation of why the vibration monitoring system (IVMS) currently relies on frequency-domain data input. What are the advantages of using frequency-domain data input compared to time-domain or time-frequency-domain methods? Moreover, considering that time-frequency analysis methods (e.g., STFT, Hilbert-Huang transform) could also be applied to this problem, the author should explain why FFT was chosen as the frequency-domain analysis method.
Response 3:
We clarified that the nuclear reactor environment typically captures data in the frequency domain, which aligns with both real-world practices and the constraints of simulation-based environments. This rationale, along with the appropriateness of using FFT in this context, has been added to Section 4.1: Dataset on page 8, starting at line 277.
Comment 4:
3.1. DTW-based Input:
The symbols used in Figure 5 (e.g., /theta_{enc} and /theta_{dec}) are not explained, and I did not see any related formula derivations or symbol definitions. To improve the readability and understanding of the figure, the author should provide further details on Figure 5 and explain the relevant symbols and their physical meanings.
Response 4:
To improve clarity, we have added detailed explanations of the symbols used in Figure 5, ensuring all relevant terms are well-defined. These updates enhance the figure’s readability and understanding.
Comment 5:
3.1. DTW-based Input:
In Figure 6, the author mentions that it shows the difference between Euclidean distance and DTW (Dynamic Time Warping) comparisons for normal and abnormal data. However, the meaning of each curve in the two subplots is unclear. The figure lacks a legend, and there is no explanation of whether different colors have specific meanings. Additionally, there is no further discussion of Figure 6 in the subsequent text, which leaves me confused about the message the author intends to convey. I recommend the author provide a more detailed explanation of Figure 6 and, if necessary, update the figure to make it more understandable. Moreover, the text in the figure is too small; I suggest increasing the font size to improve readability.
Response 5:
We appreciate your feedback on Figure 6. To address these concerns, we added a legend to clarify the meaning of each curve, increased the font size for better readability, and included additional explanations in the main text to ensure the figure’s clarity. These updates are included in Section 3.1: DTW-based Input, starting on page 7, line 210.
Comment 6:
4.1 Dataset:
There are two repetitive paragraphs in this section. Is there any specific reason for this arrangement? The author should clarify this part to help readers understand.
Response 6:
Thank you for pointing this out. We have removed the repetitive paragraphs to ensure clarity and conciseness in Section 4.1: Dataset.
Comment 7:
4.2 Experiment Setting:
The author evaluates model performance using three configurations: amplitude alone, phase alone, and a combination of both. Why were these three configurations chosen? The author needs to explain the rationale for selecting these configurations and specify whether other input features were considered for evaluating the network performance. If necessary, please add experiments to validate the impact of different input configurations on model performance.
Response 7:
We elaborated on the rationale for using amplitude, phase, and combined configurations as they represent key components of frequency-domain data. To further support this choice, we added experiments to validate the impact of these configurations on model performance. These details are included in Section 4.3.1: Comparison of Magnitude and Phase Suitability on page 10, starting at line 331.
Comment 8:
- Experiment and Results Analysis:
The author mentions training and validation with simulated data, as real data could not be obtained, as stated in the introduction. However, there is typically a large discrepancy between simulated data and real experimental signals. The author should further explain why the training results obtained with simulated data can be used for fault diagnosis of real experimental signals. If normal signal data from a real environment is available, could the author include tests on real normal signals to address concerns about the neural network trained on simulated data's ability to diagnose real signals?
Response 8:
We acknowledge this concern and have provided additional explanations about the limitations posed by the lack of access to real nuclear power plant data due to confidentiality. We also highlighted the unique contributions of our study in addressing these challenges. These updates are included in Section 5: Conclusion on page 15, starting at line 443.
Once again, thank you for your valuable comments. We believe that your feedback has significantly enhanced the quality of our manuscript.

Round 2
Reviewer 1 Report
Comments and Suggestions for Authors
This paper can be accepted.
Reviewer 2 Report
Comments and Suggestions for Authors
- The revised version of the paper is clearly improved.
- I appreciated that the authors took into account all my previous remarks
- With this, i give my notice of acceptance for the publishing of the paper as it is.